# Histone Deacetylases (HDACs): Promising Biomarkers and Potential Therapeutic Targets in Thymic Epithelial Tumors

**DOI:** 10.3390/ijms24054263

**Published:** 2023-02-21

**Authors:** Kostas Palamaris, Luisa-Maria Tzimou, Georgia Levidou, Christos Masaoutis, Irene Theochari, Dimitra Rontogianni, Stamatios Theocharis

**Affiliations:** 1First Department of Pathology, National and Kapodistrian University of Athens, 11527 Athens, Greece; 2Department of Pathology, Paracelsus Medical University, 90419 Nuremberg, Germany

**Keywords:** thymic epithelial tumors, thymic carcinoma, histone deacetylases, epigenetics, prognosis, biomarker

## Abstract

Histone deacetylases (HDACs) are core epigenetic factors, with pivotal roles in the regulation of various cellular procedures, and their deregulation is a major trait in the acquisition of malignancy properties. In this study we attempt the first comprehensive evaluation of six class I (HDAC1, HDAC2, HDAC3) and II HDACs (HDAC4, HDAC5, HDAC6) expression patterns in thymic epithelial tumors (TETs), with the aim of identifying their possible association with a number of clinicopathological parameters. Our study revealed higher positivity rates and expression levels of class I enzymes compared to class II. Sub-cellular localization and level of staining varied among the six isoforms. HDAC1 was almost exclusively restricted to the nucleus, while HDAC3 demonstrated both nuclear and cytoplasmic reactivity in the majority of examined specimens. HDAC2 expression was higher in more advanced Masaoka–Koga stages, and displayed a positive correlation with dismal prognoses. The three class II HDACs (HDAC4, HDAC5, HDAC6) exhibited similar expression patterns, with predominantly cytoplasmic staining, that was higher in epithelial rich TETs (B3, C) and more advanced tumor stages, while it was also associated with disease recurrence. Our findings could provide useful insights for the effective implementation of HDACs as biomarkers and therapeutic targets for TETs, in the setting of precision medicine.

## 1. Introduction

Thymic epithelial tumors (TETs) originate from the epithelial cells of the thymus gland, and are the most frequent neoplasms of mediastinum. They engulf a broad spectrum of histologically divergent tumor subtypes, with completely different clinical outcomes [1]. The clinicopathological heterogeneity of thymic neoplasms reflects their distinctive molecular profiles. Accumulated data of integrated genomic TETs analyses suggest that the broad histological subtypes do not represent a molecular and phenotypic continuum, but that they are distinct entities, characterized by different molecular aberrations and unique pathogenetic routes [2,3]. However, further studies are urgently needed for a more in-depth analysis of tumor molecular background and identification of signature alterations, that could also serve as novel targets for therapeutic intervention.

In this direction, a large breadth of both in vitro experimental studies and human specimens’ analyses have demonstrated the deregulation of epigenetic networks, including non-coding RNAs, DNA methylation, and histone modifications as a key driver of thymic tumorigenesis [4]. A multiplicity of noncoding RNA molecules, including micro-RNAs and long noncoding RNAs, are either upregulated or downregulated in thymic neoplasms, prompting corresponding changes to the expression profile of their target genes, which include both oncogenes and tumor-suppressors [5,6,7,8]. Abnormal DNA methylation of CpG sites, located at regulatory regions, primarily in gene promoters, is also a hallmark feature of TETs. Hypermethylated loci are predominantly tumor suppressor genes, especially cell cycle checkpoints and negative regulators of oncogenic pathways. Large-scale DNA methylation analyses have revealed different methylation profiles among distinct subtypes, with more aggressive TCs displaying a higher methylation index [9,10,11]. Histone modifications refer to the dynamic control of the 3D chromatin structure via heterogenous mechanisms, which can be segregated into two distinct categories: the reconstruction of nucleosome organization; and the modification of histone tails, by enzyme-mediated addition or removal of various chemical elements (methylation, phosphorylation, acetylation, and ubiquitination). These chemical modifications adapt genomic regions, in order to maintain silencing or activation of gene expression. Histone acetylation represents arguably the best-studied covalent histone modification, and occurs at evolutionarily conserved lysine residues of nucleosome complexes. The addition of acetyl groups loosens the chemical interactions between histones and DNA nucleotides, and facilitates the decompensation of tightly packed chromatin, rendering it more accessible to transcriptional machinery components and establishing permissive chromatin states [12]. This dynamic process is controlled by a fluctuating equilibrium between the reversible activity of two antagonizing enzyme families: histone acetyltransferases (HATs), which install acetyl groups, enabling activation of gene expression; and histone deacetylases (HDACs), that remove them, creating inhibitory histone “marks” [13,14]. HDACs are a chemically and functionally diverse group of enzymes, categorized into four classes: class I, II (a, b), III, and IV. Class I includes HDAC1, -2, -3, and -8, which are ubiquitously expressed, while class II encompasses HDAC4, -5, -6, -7, -9, and -10, which are characterized by a tissue-specific expression pattern. Both class I and II enzymes are detected in the nucleus as well as in cytoplasm and other cellular organelles. Besides their histone-mediating function, they also regulate the levels and activity of multiple other proteins, by post-translational addition of acetyl groups, that modulates their stability [14]. The remaining two HDAC classes are class III HDACs, or sirtuins (SIRT), and class IV, which consists exclusively of HDAC11 [14]. Class III and class IV isoforms are characterized by a foremost nuclear localization. 

As expected, by their critical role as epigenetic regulators, HDACs control expression of multiple genes that mediate different cellular processes, such as proliferation, apoptosis, metabolism, and immunogenicity, and their deregulation is a major trait in the acquirement of malignancy properties [15]. A large breadth of studies has evaluated the expression levels of class I and II HDACs in a wide range of tumors, attempting correlations with patients’ survival and other clinicopathological parameters. Increased levels of class I and class II isoforms have been detected via immunohistochemistry or molecular techniques in heterologous and histologically divergent epithelial, mesenchymal, and central nervous system neoplasms, as well as in melanoma [16,17,18,19,20,21,22,23,24,25,26,27,28,29,30,31,32]. The aberrant expression of HDACs in cancer, and their critical role in regulating different aspects of tumor cells’ biology, suggests that they could serve as ideal therapeutic targets. Indeed, a broad spectrum of HDAC inhibitors (HDACi) have already been developed and entered clinical trials for a wide range of tumors [33,34,35].

Considering the lack of systematic analyses of the HDAC expression profile in TETs, along with the fact that HDACs could represent novel therapeutic targets, following the trend set by other malignancies, in the present study we aim to evaluate the immunohistochemical expression of six HDAC isoforms, belonging to classes I and II, in a cohort of TETs, covering the entire spectrum of major histological subtypes. Moreover, we attempt to correlate HDACs’ expression patterns with patients’ survival and other clinicopathological parameters (Table 1).

## 2. Results

### 2.1. Expression of HDAC1 in TETs and Associations with Clinicopathological Characteristics

HDAC1 expression was observed in 96.5% of the examined cases, which primarily showed nuclear staining, and had a median H-score of 200 (range 0–300, Table 2). Cytoplasmic immunoreactivity was detected in only three specimens (3.5%) (Figure 1). There was not any significant association between WHO histological type, Masaoka–Koga stage, presence of relapse, or patients’ overall survival (OS) (*p* > 0.10, Figure 2). A positive reaction within the lymphocytic component was observed in 50 cases, all of them showing nuclear staining and only 2 displaying a cytoplasmic staining. There was not any significant correlation with the remaining parameters presented in. The expression of HDAC1 in the lymphocytic component was not correlated with any of the clinicopathological parameters.

### 2.2. Expression of HDAC2 in TETs and Associations with Clinicopathological Characteristics

HDAC2 expression was encountered in 97% of our cases, with both nuclear and cytoplasmic localization and a median H-score of 85 (range 0–300) (Table 2, Figure 1). While there was no significant correlation of H-score with WHO histological types, or the presence of relapse (*p* > 0.10), TETs with an advanced Masaoka–Koga stage (II or higher) displayed a higher HDAC2 H-score compared to stage I ones (Mann–Whitney, *p* = 0.045, median value 100 vs. 80, Figure 3). Moreover, an increased HDAC2 H-score was correlated with a worse OS (log-rank test, *p* = 0.008, Figure 3C). The associations with the remaining clinicopathological features, such as patients’ age and gender, as well as presence of relapse, were not significant. Lymphocytic immunoreactivity was observed in 49 cases and showed a positive correlation with the expression of HDAC2 in the epithelial component of TETs (Spearman correlation coefficient, R = 0.5958, *p* < 0.0001). However, lymphocytic HDAC2 levels were not correlated with any of the clinicopathological parameters. 

### 2.3. Expression of HDAC3 in TETs and Associations with Clinicopathological Characteristics

HDAC3 positive expression was observed in 94% of cases, with both nuclear and cytoplasmic localization (Table 2, Figure 1). Fifty-eight cases displayed simultaneous cytoplasmic and nuclear immunoreactivity, while three cases showed exclusively cytoplasmic and 13 displayed only nuclear positivity. In the remaining five specimens, no staining was detected. Interestingly, there was a positive correlation between nuclear and cytoplasmic H-score (Spearman correlation coefficient, R = 0.4182, *p* < 0.001). Among different WHO histological subtypes, less frequent nuclear positivity was encountered in thymic carcinomas (66.7%), compared to the rest of the TETs (92.8%) (Fisher’s exact test, *p* = 0.044, Figure 4A). No significant correlation of HDAC3 expression with Masaoka–Koga stage, the presence of relapse, or patients’ overall survival (*p* > 0.10, Figure 4B–D) was observed. There were not any significant correlations with the remaining clinicopathological parameters presented in Table 1. Regarding lymphocytic component, thirty-one cases demonstrated a nuclear positivity for HDAC3, and four showed cytoplasmic staining (two of them being only positive in the cytoplasm), without significant correlation with any of the clinicopathological parameters. 

### 2.4. Expression of HDAC4 in TETs and Associations with Clinicopathological Characteristics

HDAC4 staining, with an exclusively cytoplasmic pattern, was observed in 70% of the examined cases, with a median H-score of 45, and a range of 0–210 (Table 2, Figure 1). B2- and B3-type TETs and thymic carcinomas were more frequently positive for HDAC4 compared to the other WHO types (Fisher’s exact test, *p* = 0.03, 63% vs. 37%). Moreover, epithelial rich (namely B3 and thymic carcinoma) TETs tended to show a higher HDAC4 H-score compared to the rest of the types (Mann–Whitney, *p* = 0.061, median value 60 vs. 25, Figure 5A), but this relationship was of marginal significance. A higher HDAC4 H-score was detected in TETs with an advanced Masaoka–Koga stage (II or higher), compared to stage I cases (Mann–Whitney, *p* = 0.003, median value 50 vs. 0, Figure 5B–D). There was not any significant correlation of HDAC4 expression with the presence of relapse, or with patients’ OS or the remaining clinicopathological parameters presented in Table 1. No staining was observed in the lymphocytic component.

### 2.5. Expression of HDAC5 in TETs and Associations with Clinicopathological Characteristics

HDAC5 positivity was observed in 67% of the examined specimens (Table 2), with a primarily cytoplasmic pattern (62%) (Figure 1). Nuclear staining was observed in only 11 cases (13%), with four of them showing an exclusively nuclear immunoreactivity. Epithelial rich TETs (namely B3-type and thymic carcinomas) displayed positive cytoplasmic HDAC5 staining more often (Fisher’s exact test, *p* = 0.022, 37% vs. 13%), and a higher cytoplasmic HDAC5 H-score, compared to the rest of the WHO types (Mann–Whitney, *p* = 0.002, median value 160 vs. 20, Figure 6A). TETs with an advanced Masaoka–Koga stage (namely III/IV) showed a higher cytoplasmic HDAC5 H-score compared to the rest of the types (Mann–Whitney, *p* = 0.048, median value 120 vs. 40, Figure 6B). Furthermore, an increased HDAC5 cytoplasmic H-score was correlated with the presence of relapse (Mann–Whitney, *p* = 0.026, median value 120 vs. 10). There was not any correlation of HDAC5 expression with patients’ OS or with the remaining clinicopathological parameters. Only in two cases was there a positive cytoplasmic immunoreaction in the lymphocytic component of the tumor.

### 2.6. Expression of HDAC6 in TETs and Associations with Clinicopathological Characteristics

HDAC6 immunoreactivity was observed in 29.6% of the examined cases, and was predominantly cytoplasmic (Table 2, Figure 1). Epithelial rich TETs (namely B3-type and thymic carcinomas) displayed more frequent positivity (Fisher’s exact test, *p* = 0.017, 48% vs. 18%, Figure 7A) and higher H-scores compared to the rest of the histological types (Mann–Whitney, *p* = 0.008, median value 30 vs. 0). Moreover, a higher cytoplasmic HDAC6 H-score was encountered in tumors with an advanced Masaoka–Koga stage (IVa or IVb) (Mann–Whitney, *p* = 0.008, median value 70 vs. 0, Figure 7B). There was not any significant correlation with the presence of relapse or patients’ OS or with any of the remaining clinicopathological parameters. No expression was observed in the lymphocytic component.

## 3. Discussion

HDACs are major epigenetic factors, with crucial roles in the regulation of various cellular processes, such as cell cycle control, apoptosis, metabolism, autophagy, metastasis, and angiogenesis. The multidimensional role that their abnormal expression exerts in carcinogenesis has been thoroughly investigated in a wide range of solid and hematologic malignancies. However, only scant data exist concerning the potential role of HDACs in thymic tumorigenesis. 

Our study revealed higher positivity rates and expression levels of class I enzymes (HDAC1, HDAC2, HDAC3). Positive staining of HDAC1, HDAC2, and HDAC3 was observed in almost all examined cases, with positivity rates of 96.5%, 97%, and 94%, respectively. Concerning the expression levels of the three class I enzymes, they were highest in the case of HDAC1, which showed a median H-score of 200 (range: 0–300), with diffuse staining of moderate-to-high intensity across different WHO subtypes and tumor stages. HDAC2 expression was higher in more advanced Masaoka–Koga stages (*p* = 0.045), and positively correlated with dismal prognoses (*p* = 0.0078). A limitation of our study in this context is that survival data were available only in a small subgroup of our cohort, in which we were able to perform survival analysis, and therefore we were not able to perform multivariate survival analysis in order to provide a possible prognostic nomogram including HDAC immunoexpression. However, the positive correlation of HDAC2 levels with reduced patients’ survival, which emerged in our study, harmonizes with findings from similar studies, which have identified HDAC2 as a negative prognostic factor, associated with diminished disease-free survival (DFS) or OS in a wide range of malignancies, including oral squamous cell carcinoma [36], hepatocellular carcinoma [21], cholangiocarcinoma [37], gastric [38] and colorectal adenocarcinoma [39], and endometrioid cancer [40]. Increased HDAC2 has been also associated with advanced stage and poor tumor differentiation in gastrointestinal [39] and breast carcinomas [31]. Variations were also observed in HDAC3 staining among different WHO subtypes, as nuclear immunoreactivity was less frequent in the more aggressive thymic carcinomas, compared to the rest of the TETs categories (Fisher exact test, *p* = 0.044). Such an association of HDAC3 levels with higher tumor grade has also been observed in colorectal and mammary adenocarcinomas [39,40,41]. However, in our cohort HDAC1 did not show any significant correlation with the clinicopathological parameters, although it has been previously reported to play a significant role in several types of cancer, such as gastric and lung cancers [42]. 

The three investigated class II HDACs (namely HDAC4, HDAC5, HDAC6) exhibited a predominantly cytoplasmic staining, with HDAC4 and HDAC5 showing a higher median H-score compared to HDAC6 (HDAC4 and HDAC5 median H-score 45 in both cases, ranges: 0–210 and 0–300, respectively, vs. HDAC6 median H-score 0, range: 0–100). Moreover, the cytoplasmic H-score of all three class II HDACs was higher in epithelial rich TETs (B3, C) and increased in more advanced tumor stages, while in the case of HDAC5 it was also associated with disease recurrence. HDAC4 has previously been associated with advanced tumor stage in esophageal and nasopharyngeal carcinoma [43,44]. Accordingly, HDAC5 expression has been related with worse prognosis and the presence of metastasis in breast cancer [45]. HDAC6, which represents the most thoroughly studied class II isoform, has been correlated with a higher tumor grade in salivary gland neoplasms [46], and advanced pathological stages in oral squamous cell and thyroid carcinomas [47,48].

The subcellular distribution of class I HDACs in our cohort is in agreement with previous experimental studies, which have demonstrated an almost exclusively nuclear localization of HDAC1 and HDAC2, while HDAC3 has been detected in both the nuclear and cytoplasmic compartments. What distinguishes HDAC3 from the other two class I isoforms is the presence of a nuclear export signal, that mediates its cytoplasmic translocation, via interaction with CRM-1, a cellular nuclear-export factor [49]. Sequestering of HDAC3 in the cytoplasm is also mediated by IκBa, a cytoplasmic component of the NF-κB cascade, that is capable of binding HDAC3 via ankyrin repeats and preventing its nuclear translocation [50]. In the cytoplasm, HDAC3 modulates a number of signal transduction pathways and cell cycle regulators, via controlling their acetylation levels. Deacetylation of STAT1 and STAT3 inhibits their phosphorylation, which is required for their nuclear translocation and dimerization, respectively [51,52]. Moreover, by deacetylating different lysine residues of p65, HDAC3 differentially regulates the activity of NF-κB [53,54]. Cyclin A levels are also controlled by post-translational acetylation events, and the reduced acetylation status induced by HDAC3 has been shown to induce its proteasomal degradation [55].

Class II HDACs predominantly cytoplasmic staining reconciles with their well-established role in controlling nonhistone proteins. HDAC4 and HDAC5 shuttle between the cytoplasm and nucleus, in a bidirectional translocation between the two cellular compartments, controlled primarily by phosphorylation events. The proteins 14-3-3 have been shown to bind to phosphorylated threonine residues of the two HDAC isoforms, sequestering them in the cytoplasm [56]. HDAC4’s most thoroughly studied non-histone targets include pro-apoptotic factor p53 and hypoxia-response molecule HIF-1a, two proteins activated by DNA damage or reduced oxygen availability, respectively. Such conditions of cellular stress stimulate their translocation to the nucleus, where they serve as orchestrators of a stress-induced transcriptional response, that allows cells to adapt in detrimental circumstances. Their levels and function are tightly regulated post-translationally by multiple covalent modifications, including acetylation, which can either increase their stability and functional integrity or direct them for proteasomal degradation. In the case of p53, an increased acetylation index prompts its nuclear import, enabling it to carry-out its tumor-suppressive role, while HDAC4-mediated deacetylation prevents its translocation and leads to destabilization and proteasomal degradation [57]. On the other hand, HIF1α is destabilized by acetylation, and its deacetylation by HDAC4 maintains higher protein levels and allows it to execute its tumor-promoting functions [58]. HDAC6 is characterized by predominantly cytoplasmic localization, with its main tasks being the regulation of β-catenin. Acetylation of lysine 49 of β-catenin, promotes its degradation. Therefore, HDAC6 inactivation inhibits nuclear translocation of β-catenin, resulting in the inhibition of cellular growth [59]. 

The multipronged role of HDACs in tumorigenesis suggests that they can serve as novel targets of therapeutic interventions. A number of both in vitro and in vivo studies have demonstrated the effects of HDAC inhibition in tumor cells [60]. Either as monotherapy, or in combination schemes with traditional chemotherapeutic agents, they have demonstrated the capacity to block cell proliferation, induce programmed cell death, and target the metabolic dependencies of cancer cells, that derive from the broad reprogramming of metabolic pathways. Moreover, HDAC inhibition has the potential to suppress tumor cells’ capacity to fluctuate between different phenotypes, especially the reversible transdifferentiation from an epithelial to a more mesenchymal state, achieved via activation of epithelial-to-mesenchymal transition, which enhances their invasive and metastatic capacity [60]. 

In vitro studies of HDAC inhibition provide evidence about the therapeutic capability of such HDACi, while clarifying their effect on molecular alterations and deregulation of intracellular signaling pathways, that takes place as a result of the broad epigenetic rewiring induced by HDAC [60]. In thymic neoplasms, only scarce data are available regarding the potential use of HDACi as a therapeutic strategy. Administration of the class I and IIa HDACi, valproic acid, in TC cell line TC1889, induced a G1-phase growth arrest of tumor cells, accompanied by induction of the cell-cycle negative regulator p21 [61]. Evaluation of two other HDACi, belinostat and panobinostat, on a primary cell line derived from a patient metastatic lesion, demonstrated some promising results in reducing tumor cells’ growth and proliferation [62]. Two phase II clinical trials, testing the clinical efficacy of the HDACi belinostat in two relatively large patient cohorts (41 and 26 patients, respectively), demonstrated benefits in a substantial proportion of cases. The first study included 25 patients with B-type thymoma (B1, B2, B3) and 16 with TC. All of them had received a median of 2 (range: 1–10) previous chemotherapy schemes [63]. Among them, only two patients with thymoma experienced partial response, while 25 patients achieved disease stabilization. In the second one, total enrollment consisted of 12 patients with thymoma and 14 with TC, and treatment outcomes were much more promising, with 61% response rates in thymoma patients and 21% in TC cases. In all responders, response was partial, and it was associated with a reprogramming of the immunosuppressive tumor microenvironment via a decrease in T-regulatory cells and exhausted CD8 T-cells [64]. Thus, the results of the limited number of clinical studies testing HDACi in TETs, are in complete harmony with our findings, as more aggressive TETs types, which in general show higher levels of specific HDAC isoforms, seem to be more responsive to their inhibition. Further clinical trials however are required, so that more reliable conclusions can be extracted regarding the potential of such a therapeutic strategy in this rare tumor family. 

## 4. Materials and Methods

### 4.1. Patients’ Characteristics

This is a study of archival formalin-fixed paraffin-embedded (FFPE) tissue from 91 patients with TETs, resected between 2009 and 2019, retrieved from the pathology laboratory archives of the Evangelismos General Hospital, Athens, Greece, for whom medical records were available. Patient characteristics are shown in Table 1. Thirty-nine of the patients were men (43%) and 52 women (57%), with a median age at diagnosis of 62 years (range 27–88 years). Tumors had been classified, according to the WHO classification scheme, to one of the following seven subtypes: A, AB, B1, B2, B3, micronodular thymoma with lymphoid stroma (MNT), and thymic carcinoma [65]. Their pathological staging was based on the Masaoka–Koga system, which classifies tumors into four stages, according to the invasion of thymic capsule or adjacent organs, the presence of pleural or pericardial implants, and the hematogenous/lymphogenous metastasis to extrathoracic sites [66]. The frequency of WHO subtypes was as follows: type-A 13.2%; type-AB 20.8%; type-B1 15.4%; type-B2 20.8%; type-B3 15.4%; MNT 2.2%; thymic carcinoma 12.1%. Moreover, the Masaoka–Koga stages of the patients were: I 16%; IIa 39.5%; IIb 17.3%; III 19.7%; IVa 3.5%; IVb 3.7%. The TNM staging system was not available until late in our cohort, and therefore is not included in our study. Co-existing myasthenia gravis was diagnosed in 59.3% of patients, two of whom also suffered from pemphigus vulgaris and autoimmune thyroidopathy. Chemotherapy was given to 28%, and radiotherapy to 50%, of patients for whom respective information was available; six of these patients received both chemo- and radiotherapy. Follow-up information was available for 40 patients, ranging from 5 to 134 months (median: 32 months). 

### 4.2. Immunohistochemistry

Immunohistochemistry was carried out using standard procedures in the eight TMAs. Immunostainings for HDAC1, -2, -3, -4, -5, and -6 were performed on individual formalin-fixed, paraffin-embedded tissue sections, using rabbit polyclonal anti-HDAC1 (H-51, sc-7872, Santa Cruz Biotechnology, Santa Cruz, CA, USA) and anti-HDAC2 (H-54, sc-7899, Santa Cruz Biotechnology) IgG antibodies, and mouse monoclonal anti-HDAC4 (A-4, sc-46672, Santa Cruz Biotechnology) IgG2b and anti-HDAC6 (D11, sc-28386, Santa Cruz Biotechnology) IgG2a antibodies. Antigen retrieval was performed at pH 6. The Envision (Dako, Agilent, Santa Clara, CA, USA) visualization system was used. DAB (3,3-diaminobenzidine) was used as a chromogen, and hematoxylin as a counterstain. Appropriate positive controls, according to the manufacturer, were used. As a negative control, the omitted primary antibody and substitution with an irrelevant antiserum was used.

For the purposes of the immunohistochemical evaluation, we calculated the H-score, which serves as a semiquantitative measure of the immunohistochemical protein expression levels. To calculate the H-score, the semiquantitative staining intensity score (score 1 to 3) is multiplied by the percentage of positive cells. Therefore, H-score values range between 0 and 300. The epithelial and lymphocytic components, as well as the nuclear and cytoplasmic positivity, were separately evaluated.

### 4.3. Statistical Analysis

Statistical analysis was performed by an MSc biostatistician (GL). The association between the IHC expression of HDAC1, -2, -3, -4, -5, and -6 with clinicopathological characteristics was examined using nonparametric tests, with correction for multiple comparisons, as appropriate. Survival analysis was performed using Kaplan–Meier survival curves, and the differences between the curves were compared with the log-rank test. Numerical variables were categorized according to the median value. A *p*-value of <0.05 was considered statistically significant. The analysis was performed with the statistical package STATA 11.0/SE (College Station, TX, USA) for Windows.

## 5. Conclusions

In this study, we provide the first comprehensive evaluation of class I and class II HDAC isoforms expression profiles in a large cohort of TETs. Interesting results were extracted regarding the levels and the subcellular localization of different HDACs among, WHO histological types, Masaoka–Koga pathological stages, as well as their association with patients’ prognoses. Thus, the precise identification of their expression patterns could help in the more effective deployment of HDACs as determinants of patients’ survival, and predictors of their response to HDAC-targeting agents, in the context of precision medicine.

## Figures and Tables

**Figure 1 ijms-24-04263-f001:**
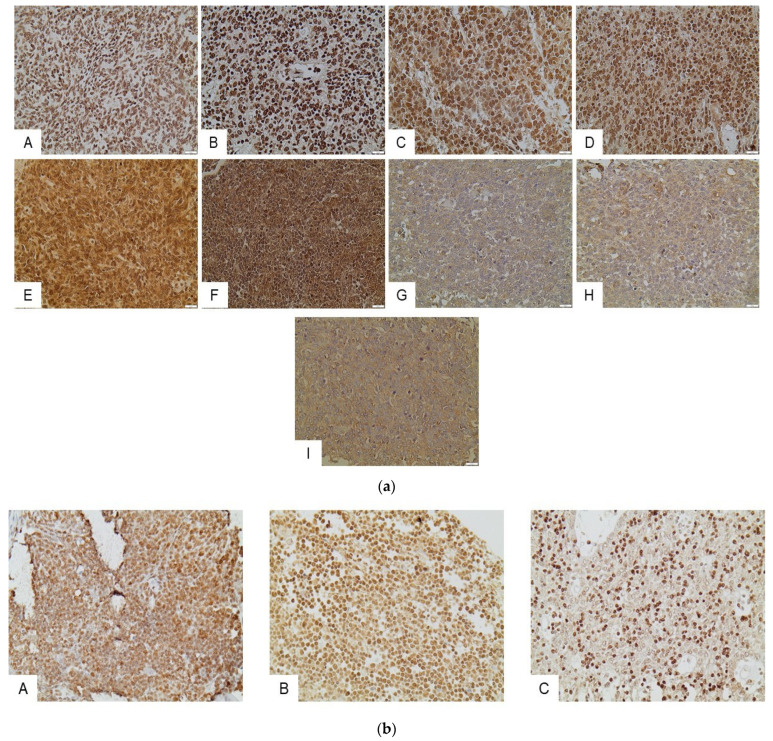
(**a**) Immunohistochemical expression of HDACs in the epithelial component of TETs. HDAC1 in an A type thymoma (nuclear) (A), and in a thymic carcinoma (nuclear) (B). HDAC2 in a thymic carcinoma (nuclear and cytoplasmic) (C), and in a B3 type thymoma (nuclear and cytoplasmic) (D). HDAC3 in a thymic carcinoma (nuclear and cytoplasmic) (E), and in a thymic carcinoma (nuclear and cytoplasmic) (F). HDAC4 in a type A thymoma (cytoplasmic). (G) HDAC5 in a type A thymoma (cytoplasmic) (H), and HDAC6 in a thymic carcinoma (I). All pictures are at ×400 magnification. (**b**) Immunohistochemical expression of HDACs in the lymphocytic component of TETs. HDAC1 in a B1 type thymoma (nuclear) (A). HDAC2 expression in an A type thymoma (nuclear) (B), and HDAC3 expression in a B2 type thymoma (nuclear) (C). All pictures are at ×400 magnification.

**Figure 2 ijms-24-04263-f002:**
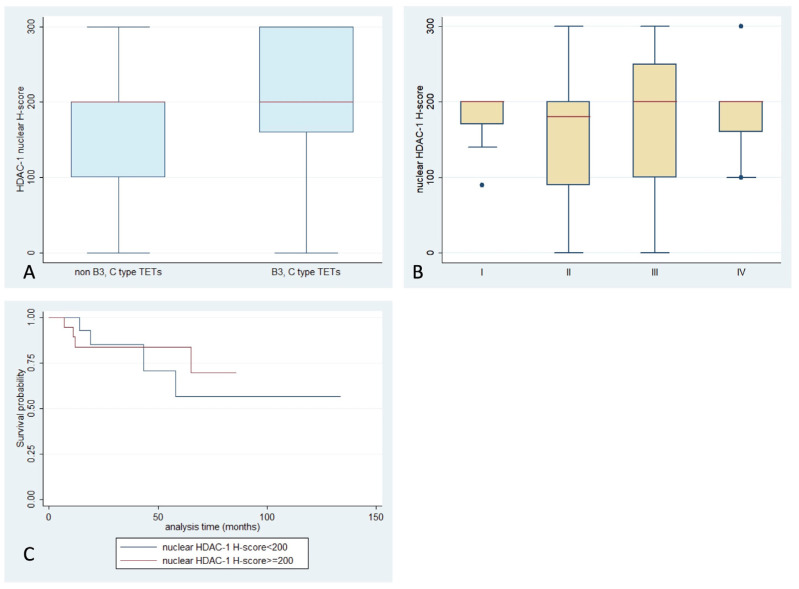
Schematic representation of the associations between nuclear HDAC1 H-score and (**A**) WHO histological type, (**B**) Masaoka–Koga stage, and (**C**) patients’ overall survival.

**Figure 3 ijms-24-04263-f003:**
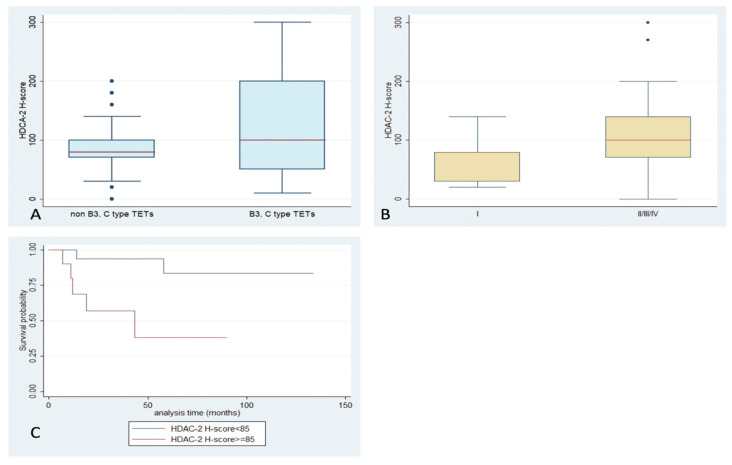
Schematic representation of the associations between HDAC2 H-score and (**A**) WHO histological type, (**B**) Masaoka–Koga stage, and (**C**) patients’ overall survival.

**Figure 4 ijms-24-04263-f004:**
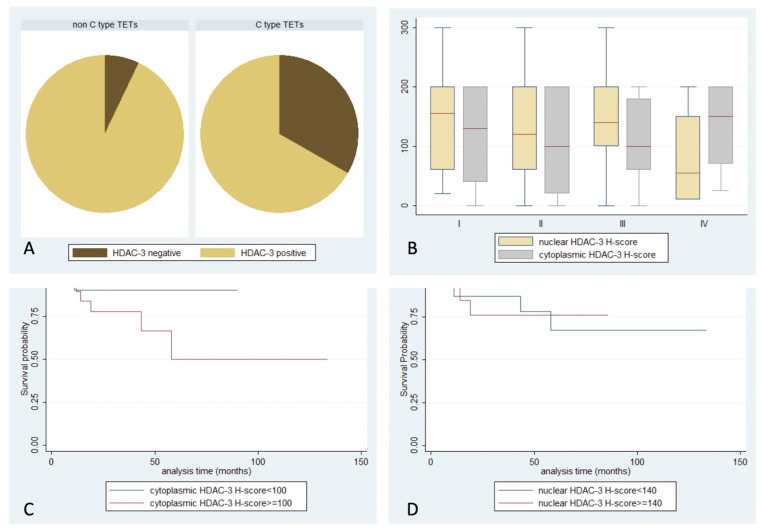
Schematic representation of the associations between (**A**) nuclear HDAC3 positivity and WHO histological type, (**B**) nuclear or cytoplasmic HDAC3 H-score and Masaoka–Koga stage, (**C**) cytoplasmic HDAC3 H-score and patients’ overall survival, and (**D**) nuclear HDAC3 H-score and patients’ overall survival.

**Figure 5 ijms-24-04263-f005:**
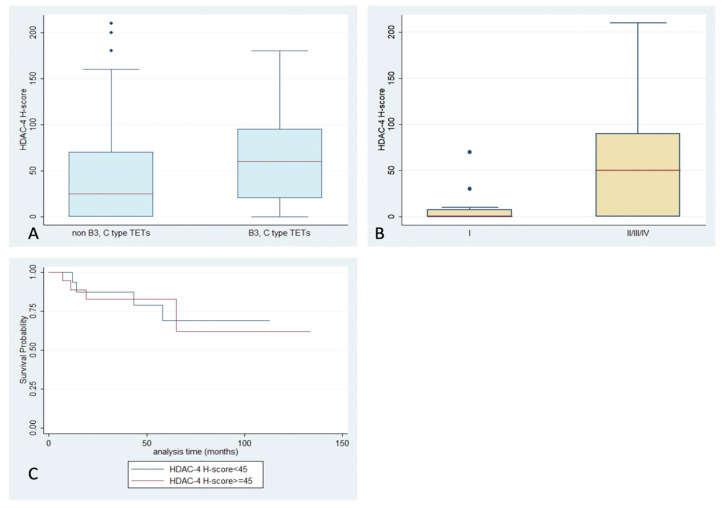
Schematic representation of the associations between HDAC4 H-score and (**A**) WHO histological type, (**B**) Masaoka–Koga stage, and (**C**) patients’ overall survival.

**Figure 6 ijms-24-04263-f006:**
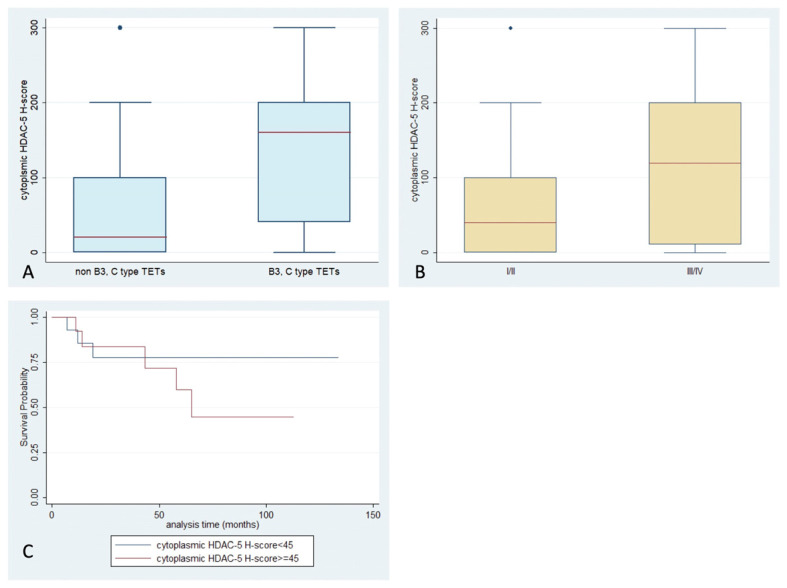
Schematic representation of the associations between cytoplasmic HDAC5 H-score and (**A**) WHO histological type, (**B**) Masaoka–Koga stage, and (**C**) patients’ overall survival.

**Figure 7 ijms-24-04263-f007:**
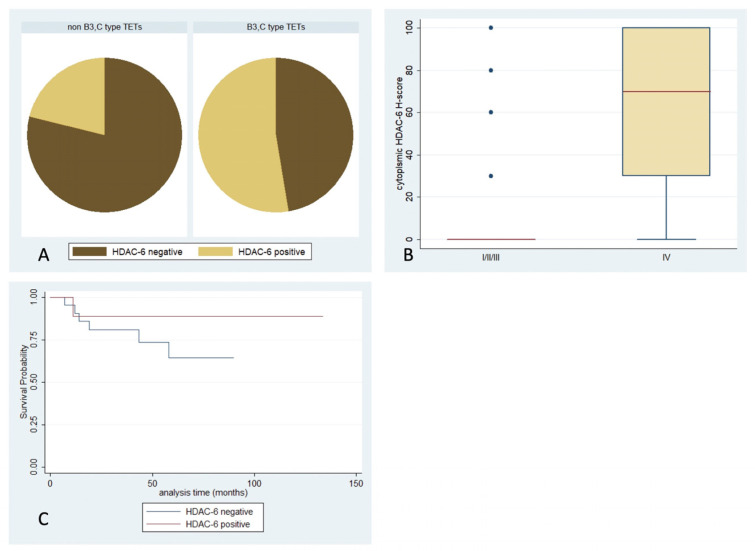
Schematic representation of the associations between cytoplasmic HDAC6 H-score and (**A**) WHO histological type, (**B**) Masaoka–Koga stage, and (**C**) patients’ overall survival.

**Table 1 ijms-24-04263-t001:** Clinicopathological characteristics of 91 patients with TETs.

Parameter	Median	Range
**Age**	62	27–88 years
	**Number**	**%**
**Gender**		
*Male*	39/91	43%
*Female*	52/91	57%
**WHO subtypes**		
Type A	12/91	13.6%
Type AB	19/91	20.8%
Type B1	14/91	15.4%
Type B2	19/91	20.8%
Type B3	14/91	15.4%
Micronodular with lymphoid stroma	2/91	2.2%
Thymic carcinoma	11/91	12.1%
**Masaoka–Koga stage**		
I	13/81	16%
IIa	32/81	39.5%
IIb	14/81	17.3%
III	64/81	19.7%
IVa	3/81	3.7%
IVb	3/81	3.7%
**Presence of myasthenia gravis**	35/59	59.3%
**Presence of chemotherapy**	11/39	28%
**Presence of radiotherapy**	19/38	50%
**Event**		
*Alive*	29/40. follow-up 5–134 months	72.5%
*Dead of disease*	11/40. within 7–65 months	27.5%
**Presence of relapse**	4/35. within 58–65 months	11%

**Table 2 ijms-24-04263-t002:** Expression of HDAC1, -2, -3, -4, -5, and -6 in the epithelial component of TETs.

	Positivity Rate	H-Score, Median	H-Score, Range
HDAC1 nuclear expression	96.5%	200	0–300
HDAC1 cytoplasmic expression	3.5%	0	0–40
HDAC2 nuclear/cytoplasmic expression	97%	85	0–300
HDAC3 nuclear expression	94%	140	0–300
HDAC3 cytoplasmic expression	77%	100	0–200
HDAC4 cytoplasmic expression	70%	45	0–210
HDAC5 nuclear expression	13%	0	0–20
HDAC5 cytoplasmic expression	62%	45	0–300
HDAC6 cytoplasmic expression	29.6%	0	0–100

## Data Availability

The data presented in this study are available on request from the corresponding author.

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
