# Peer review of "Histone Deacetylases (HDACs): Promising Biomarkers and Potential Therapeutic Targets in Thymic Epithelial Tumors"

_ijms, 2023, doi:10.3390/ijms24054263_

Round 1

Reviewer 1 Report

The Authors performed an immunohistochemical study on the expression of 6 components of the HDAC(1-6) epigenetic regulator family  in a retrospective series of 91 Thymic epithelial tumors (TET), and discuss the relevance of this family of epigenetic factors in TET biology, with notes on other systems. The paper is accurate and well written,  the results refer to the immunohistochemical findings, the literature is well cited. An interesting paper to take in mind. I found some weakness only in the fact that subcellular localization of these factors is decided and discussed, but only one figure, of medium magnification (anywhy the magnification should be cited), reports the morphological features discussed. I would like to see more figures with a higher magnification, showing the Nuclear and Cytoplasmic staining, in turn, of the different components. Please document also the lymphocytic staining (Nuclear and Cytoplasmic) (with magnification).

Moreover the Authors should put order in the WHO classification: it seems that they use indifferently the 1999 WHO classification (where the type C was included) and the 2015 WHO classification ( they cite the paper preliminary to the 2015 WHO classification). In both the 2004 and 2015 classification the type C was eliminated and substituted with the words Thymic carcinoma. Then they use in the secon part of the paper TC instead of Type C. Anyway I am providing a list of all (I hope) the linie where it is necessary to make order and to leave out type C. It would be better to cite at least the 2015 classification (now we have also the 2021 classification) and uniform the tables and the citations of histotypes.

Similarly, the Authors use the Masaoka-Koga staging system, but they do not cite neither the references (1 for Masaoka and 1 for M-Koga). There is a complete silence about the TNM staging, appeared in 2017. The Authors should state that the results were correlated with MK and that the TNM system was not available until late in their series

Please, eliminate C, substitute with Thymic carcinoma:

Pg3, linie 103 and linie 104, pg 6, linie 155 and 157, pg 8, linie 186, pg9, linie 230: eliminate type C thymoma: it is Thymic carcinoma

Last page linie 357: Masaoka not Masaok

Pg11, linie 307:....seem to be more responsive TO their inhibition

Pg11  Linie 315:Thirity not correct, use thirty

Add in materials and Methods (2009-2019) WHO classification and MK staging.

Figures: Figure 1 add magnification; please add figure(s) of high magnification showing HDAC3 in both nuclear and cytoplasmic compartment.  Show lymphocytic staining  (nuclear and cytoplasmic) with high magnification.

Reviewer 2 Report

In this study, Panamaris et al. concluded that Histone Deacetylases are promising biomarkers and potential therapeutic targets in thymic epithelial tumors. The overall study is well-designed. However, several issues remain to be addressed.

1. Please conduct the statistical analysis of HDAC levels regarding different clinical factors in Table 1.

2. Could a prognostic nomogram be constructed based on the clinical factors, including HDAC levels?

3. The IHC results in Figure 1 are not so ideal. It seems that there is some non-specific staining as the background is so dark.
